# MAGI-ACMG: Algorithm for the Classification of Variants According to ACMG and ACGS Recommendations

**DOI:** 10.3390/genes14081600

**Published:** 2023-08-08

**Authors:** Francesca Cristofoli, Muharrem Daja, Paolo Enrico Maltese, Giulia Guerri, Benedetta Tanzi, Roberta Miotto, Gabriele Bonetti, Jan Miertus, Pietro Chiurazzi, Liborio Stuppia, Valentina Gatta, Stefano Cecchin, Matteo Bertelli, Giuseppe Marceddu

**Affiliations:** 1MAGI EUREGIO, 39100 Bolzano, Italymatteo.bertelli@assomagi.org (M.B.); giuseppe.marceddu@assomagi.org (G.M.); 2MAGI’S LAB, 38068 Rovereto, Italylaboratorio@assomagi.org (S.C.); 3Istituto di Medicina Genomica, Università Cattolica del Sacro Cuore, 00168 Rome, Italy; 4UOC Genetica Medica, Fondazione Policlinico Universitario “A. Gemelli” IRCCS, 00168 Rome, Italy; 5Department of Psychological Health and Territorial Sciences, School of Medicine and Health Sciences, “G. d’Annunzio” University of Chieti-Pescara, 66100 Chieti, Italy; liborio.stuppia@unich.it (L.S.); valentina.gatta@unich.it (V.G.); 6Unit of Molecular Genetics, Center for Advanced Studies and Technology (CAST), “G. d’Annunzio” University of Chieti-Pescara, 66100 Chieti, Italy; 7MAGISNAT, Atlanta Tech Park, 107 Technology Parkway, Peachtree Corners, GA 30092, USA

**Keywords:** MAGI-ACMG, P_POT, VUS, VarSome

## Abstract

We have developed MAGI-ACMG, a classification algorithm that allows the classification of sequencing variants (single nucleotide or small indels) according to the recommendations of the American College of Medical Genetics (ACMG) and the Association for Clinical Genomic Science (ACGS). The MAGI-ACMG classification algorithm uses information retrieved through the VarSome Application Programming Interface (API), integrates the AutoPVS1 tool in order to evaluate more precisely the attribution of the PVS1 criterion, and performs the customized assignment of specific criteria. In addition, we propose a sub-classification scheme for variants of uncertain significance (VUS) according to their proximity either towards the “likely pathogenic” or “likely benign” classes. We also conceived a pathogenicity potential criterion (P_POT) as a proxy for segregation criteria that might be added to a VUS after posterior testing, thus allowing it to upgrade its clinical significance in a diagnostic reporting setting. Finally, we have developed a user-friendly web application based on the MAGI-ACMG algorithm, available to geneticists for variant interpretation.

## 1. Introduction

Classification of germline variants from genetic testing of patients with rare disorders has significantly evolved in the last decade. In the original American College of Medical Genetics guidelines (ACMG) [1], definitions of criteria for variant interpretation were quite broad, but in the following years ad hoc specifications were released for many of them to provide in depth guidance on their assignment (e.g., PVS1 [2], PS3/BS3 [3], PP3/BP4 [4]).

In addition, gene- or disease-specific guidelines have been developed by expert panels, to evaluate accurately gene-specific features, disease prevalence, and inheritance patterns, or characteristic disease-causing mechanisms [5,6].

Currently, a few variant curation tools are available online to help the standardized interpretation of germline variants, such as VarSome [7], Franklin (https://franklin.genoox.com, accessed on 1 August 2023—Franklin by Genoox), InterVar [8], and the ClinGen Variant Curation Interface [9].

Although there has been a substantial effort to develop standardized guidelines and protocols in the field of variant interpretation, each of the aforementioned tools employs different definitions and cutoffs to assign the classification criteria, relying on internal calibration of thresholds, database accessibility and users’ contribution. Therefore, the interpretation of the same variant may vary using different tools, often shifting between the three intermediate classes, likely benign/uncertain significance/likely pathogenic. Moreover, many databases used to retrieve variant statistics and information are constantly updated, and new functional evidence and case reports become available in the literature. In addition, many laboratories are specialized in the analysis of particular macro-areas of rare diseases, and therefore internal databases represent valuable sources of knowledge acquired through the analysis of several affected individuals, revealing, for instance, sub-population specific causative variants. Thus, as knowledge evolves, re-evaluation of dubious variants in time is encouraged, as the initial interpretation might need modifications.

We have previously described the integration of the VarSome Application Programming Interface (API) into the NGS data analysis workflow of our molecular genetics’ laboratory [10], followed by the development of an interactive ACMG-based classifier that allows us to interpret single nucleotide variants (SNVs) or small deletions/insertions (indels) from NGS-based testing, using the information retrieved through the VarSome environment [7] and which performs variant classification according to the recommended ACMG combinatory rules, with some internally defined modifications [11].

According to the ACMG recommendations [1], the Association for Clinical Genomic Science (ACGS) suggested that a sub-classification system for variants of unknown significance (VUS) might be useful for laboratories to decide which of these should be reported, according to the different levels of evidence supporting their pathogenicity, and according to the likelihood that further data might allow a reclassification of variants as likely pathogenic or pathogenic [12]. Indeed, retrospective testing, such as appropriate familiar segregation analysis, might be useful to demonstrate the de novo occurrence of a VUS in a gene associated with a dominantly inherited disorder (PS2/PM6 criteria), or the cosegregation of a VUS in multiple affected individuals separated by a significant number of meioses (PP1 criterion), or the in-trans occurrence of the VUS with a likely pathogenic/pathogenic variant in the same gene for disorders inherited in a recessive manner (PM3 criterion).

In this work, we propose a subclassification scheme for VUS, to automatize the selection of which variants should be reported in the diagnostic setting according to their different proximity either to the likely pathogenic/pathogenic classes or to the likely benign/benign ones. Moreover, taking into consideration that further segregation analysis might allow us to add specific criteria to the VUS and upgrade its classification, a pathogenicity potential criterion (P_POT) is automatically added under specific circumstances to highlight VUS that should be included in the final diagnostic report.

In the developed interpretative algorithm, the attribution of some ACMG criteria is customized by comparison with VarSome implementation, while others are kept as attributed by VarSome [11]. We have also developed MAGI-ACMG, a web application that offers geneticists a variant interpretation tool that applies the described framework and can be accessed through a web browser (http://magiacmg.magiclinici.it:8805, accessed on 1 August 2023).

## 2. Materials and Methods

### 2.1. MAGI-ACMG Algorithm Description

The development of an automated tool for the classification of sequence variants according to the original combinatorial ACMG rules, the integration of VarSome API in our diagnostic pipeline and the customization of some ACMG criteria have been described before [10,11]. Briefly, all variants with a decision MAF (minor allele frequency) below 3%—calculated by integrating frequencies from dbNSFP, VEP and gnomAD—are submitted for annotation through the VarSome Stable-API environment [11]. The attribution of a number of criteria is customized through the MAGI-ACMG algorithm, and is therefore independent of the VarSome specifications. The final classification is reached through the combinatorial scheme proposed by the ACMG guidelines 2015.

The MAGI-ACMG algorithm performs the following Strength modifications to:

PM2: if the criterion is triggered at a Supporting or Strong level, this algorithm re-assigns the standard Moderate intensity;BP6: if the criterion is triggered at a Very Strong or Moderate level, the algorithm re-assigns a Supporting intensity. A subsequent strength confirmation is afterwards performed by the geneticist to confirm or upgrade the criteria to a Strong level.

The MAGI-ACMG algorithm independently evaluates the following criteria:

PVS1: this criterion is assigned using the AutoPVS1 tool [13], which is based upon the detailed ClinGen guidelines for the application of the PVS1 criteria [2].PP3:
-For missense variants, at least 2/3 of the following must be true: (1) REVEL score ≥ 0.644; (2) CADD (score ≥ 25.3); (3) 8/15 of the rank scores of the 15 funtional predictors (DANN, EigenPC, FATHMM, LRT, MCAP, MetaLR, MetaSVM, MutPred, MutationAssessor, MutationTaster, PROVEAN, Polyphen2HDIV, Polyphen2HVAR, SIFT, VEST3) ≥ 0.644;-For missense and splicing variants: calculated as for missense variants, if not applicable assigned if AdaBoost score ≥ 0.708 AND RF score ≥ 0.515;-For synonymous and splicing variants: assigned if AdaBoost score ≥ 0.708 AND RF score ≥ 0.515;-For intronic variants (±5): assigned if AdaBoost score ≥ 0.708 AND RF score ≥ 0.515.BP4:
-For missense variants, at least 2/3 of the following: REVEL score ≤ 0.29; CADD score ≤ 22.7; 8/15 of the rank scores of the 15 funtional predictors (DANN, EigenPC, FATHMM, LRT, MCAP, MetaLR, MetaSVM, MutPred, MutationAssessor, MutationTaster, PROVEAN, Polyphen2HDIV, Polyphen2HVAR, SIFT, VEST3) ≤ 0.29;-For missense and splicing variants: calculated as for missense variants, if not applicable it is assigned if AdaBoost score ≤ 0.708 and RF score ≤ 0.515;-Intronic and splicing (a ± 5): assigned if AdaBoost score ≤ 0.708 and RF score ≤ 0.515.

REVEL and CADD optimal cutoffs were retrieved from the recent ClinGen calibration study for computational predictors [4]. The REVEL score cutoffs (0.644 and 0.29) were applied also to the other predictors rank scores retrieved from dbNSFP v.4 [14]. AdaBoost and RF score cutoffs were retrieved from [15].

### 2.2. Subclassification of Variants of Uncertain Significance

According to the recommendations of ACGS [12], we have subclassified VUS into three categories, according to their different proximity to the upper (likely pathogenic and pathogenic) or lower (likely benign and benign) classes: Hot, Middle and Cold (Table 1).

In the diagnostic setting, it is important to report only VUS that have a high chance of being re-evaluated as likely pathogenic following posterior testing, such as familial segregation analysis that might reveal: (1) the de novo occurrence of a VUS in a gene associated with a dominantly inherited disorder; (2) co-segregation of a VUS in a significant number of affected family members (PP1); and (3) genes associated with recessive disorders, the in-trans configuration of a VUS with a pathogenic/likely pathogenic variant in the same gene (PM3).

In order to select which variants should be reported in the diagnostic setting, we introduced a “pathogenicity potential” criterion (P_POT), which is assigned by the algorithm to Middle VUS in the following conditions, at a Moderate level:If in the same individual another pathogenic/likely pathogenic/Hot/Middle VUS is identified in a gene associated with an autosomal recessive disorder;If the middle VUS is in the homozygous state in a gene associated with autosomal recessive disorder;If the middle VUS is in the heterozygous state in a gene associated with an autosomal dominant disorder;If the middle VUS is the hemizygous state in a gene associated with an X-linked condition.

The P_POT criterion is downgraded to a Supporting level if the Middle VUS has the following criteria combinations:1 PM criterion + 2 PP criteria;2 PM criteria.

In these conditions, Middle VUS receiving a P_POT criterion become potentially Hot and can be included in the clinical report. When familial segregation is already available (contextual trio analysis or parallel analysis of multiple affected individuals), the P_POT can be inactivated and substituted by the corresponding pathogenicity criterion, according to the different situations described above. Contextual analysis might also demonstrate that the pathogenicity criteria are not applicable, and the opposite benignity criteria can be applied instead (BS4, BP2, BS2). Therefore, in the diagnostic setting both Cold and Middle VUS are not reported.

### 2.3. Implementation of MAGI ACMG Web Application

The front end of the web application is built using Vue.Js. In this part, the user uploads the variants either manually in the “Input DATA” text area or by uploading a *.csv* file using the format SAMPLEID, PANEL_NAME, ANNOTATION. The SAMPLEID and PANEL can be any string of text, while the ANNOTATION should be GENE_NAME:NM#:c.123A>T (e.g., CELSR1:NM_014246.4:c.5165G>A). After the CALCULATE button is pressed, the analysis begins, and upon completion, the list of uploaded variants is displayed in a lateral section. By selecting the queried variants, the MAGI-ACMG checklist is displayed with the criteria automatically assigned. The user can interact with the checklist by turning on other appropriate criteria, and the final verdict is recalculated. A PDF file can also be generated containing variant information such as annotation, assigned criteria, interpretation, and a list of functional predictors scores (Figure 1).

The ACMG MAGI web application is available at http://magiacmg.magiclinici.it:8805, accessed on 1 August 2023. The source code of the application can be found at the following link: https://gitlab.com/magieuregio/magi-acmg, accessed on 1 August 2023.

## 3. Results

### 3.1. In Silico Predictors, Application of Criteria BP4/PP3

For this variant in the *SERPINA6* gene [NM_001756.4:c.1165G>A, NP_001747.3:p.(Asp389Asn)], neither BP4 nor PP3 can been applied by the MAGI-ACMG algorithm because although 11/15 predictors are damaging, REVEL score: 0.518 (>0.29 but <0.644) and CADD score is 24.78 (>22.7 but <25.3). The PM2 criterion is always kept at a Moderate level (Table 2, Figure 2).

### 3.2. AutoPVS1 Implementation

For the variant NM_000372.5:c.1586del, NP_000363.1:p.(Leu529Tyrfs*7) in the *TYR* gene, the PVS1 criterion has been assigned at a Moderate level using the AutoPVS1 tool according to the specific guidelines [2]. The PM2 criterion is always kept at a Moderate level (Table 3, Figure 3).

### 3.3. Segregation Analysis: P_POT (for In-Trans Allelic Configuration)

For the variant NM_206933.4:c.5213T>C, NP_996816.3:p.(Phe1738Ser) in the *USH2A* gene, PP3 is assigned because 14/15 predictors are damaging, Revel score: 0.767 (>0.644) and CADD score: 28.30 (>25.3). Moreover, this VUS was found in an individual clinically diagnosed with Usher syndrome (PP4) who also carried a pathogenic frameshift variant in the same gene (*USH2A*:NM_206933.4:c.2299del), but with unknown phase. Therefore, the pipeline automatically introduced the P_POT criterion at Supporting strength to highlight that segregation analysis might reveal in-trans configuration, thus allowing us to classify the variant as likely pathogenic (Table 4, Figure 4).

### 3.4. Segregation Analysis: P_POT (for De Novo Occurrence)

For the missense variant in the *CELSR1* gene *NM_014246.4:ex9:c.5165G>A:NP_055061.1:p.(Arg1722Gln)* a P_POT criterion is automatically assigned at a Moderate level to highlight the possibility that the subsequent segregation analysis in the proband’s parents might reveal the de novo occurrence of the variant, as pathogenic variants in the *CELSR1* gene are associated with lymphatic malformation 9 (OMIM #619319), which is inherited in an autosomal dominant fashion. The PP3 criterion is assigned because 13/15 predictors indicate a damaging effect of the variant, Revel score: 0.723 (>0.644) and CADD score: 29.1 (>25.3) (Table 5, Figure 5).

## 4. Discussion

The purpose of medical genetic testing is to report only variants that might be relevant for the molecular diagnosis of the condition for which the patient has been referred to the clinician. Indeed, it is important to avoid the disclosure of information that might lead to an incorrect diagnosis or be misunderstood by both clinicians and patients. Reporting of any type of VUS might also lead to costly, time consuming, and often inconclusive cascade familial testing. Therefore, we support the importance of concurrent familial testing that allows the contextual evaluation of segregation data, enabling a faster and more accurate answer to the clinical question.

We propose a subclassification scheme for VUS, and introduce the automatic assignment of a pathogenicity potential criterion (P_POT) to the so-called Middle VUS as a proxy for posterior criteria that might allow us to upgrade their clinical significance. The criterion is automatically assigned by the MAGI-ACMG algorithm, according to the inheritance pattern of the disorder associated with the gene in which the variant is found, or in the case of the concurrent presence of two variants, in the same gene associated with a recessive disorder. As can be seen in Appendix A, the MAGI-ACMG classifier performs in the same way as VarSome when evaluating P, LP and VUS variants. However, the MAGI-ACMG algorithm allows VUS variants to be subclassified into the three categories Cold, Middle and Hot.

We have focused on segregation criteria, and did not take into consideration the automation of the P_POT criterion for other types of information, such as functional testing (PS3) or disease specificity (PP4). The latter might be difficult to be implemented in an automatic system which is not built for taking into consideration phenotypic information, especially considering that in our laboratories a great proportion of tests are performed for genetically heterogeneous conditions such as retinitis pigmentosa or inherited retinal disorders.

## 5. Conclusions

This work represents an attempt to introduce into the diagnostic setting a VUS subclassification, as proposed by ACMG and ACGS guidelines. This approach will assist geneticists in identifying those VUS that have a high chance of becoming likely pathogenic or pathogenic following posterior segregation testing by using a P-POT proxy criterion. We have also implemented MAGI-ACMG, a web application that is available to geneticists to interpret their variants of interest, based on the MAGI-ACMG custom algorithm.

## Figures and Tables

**Figure 1 genes-14-01600-f001:**
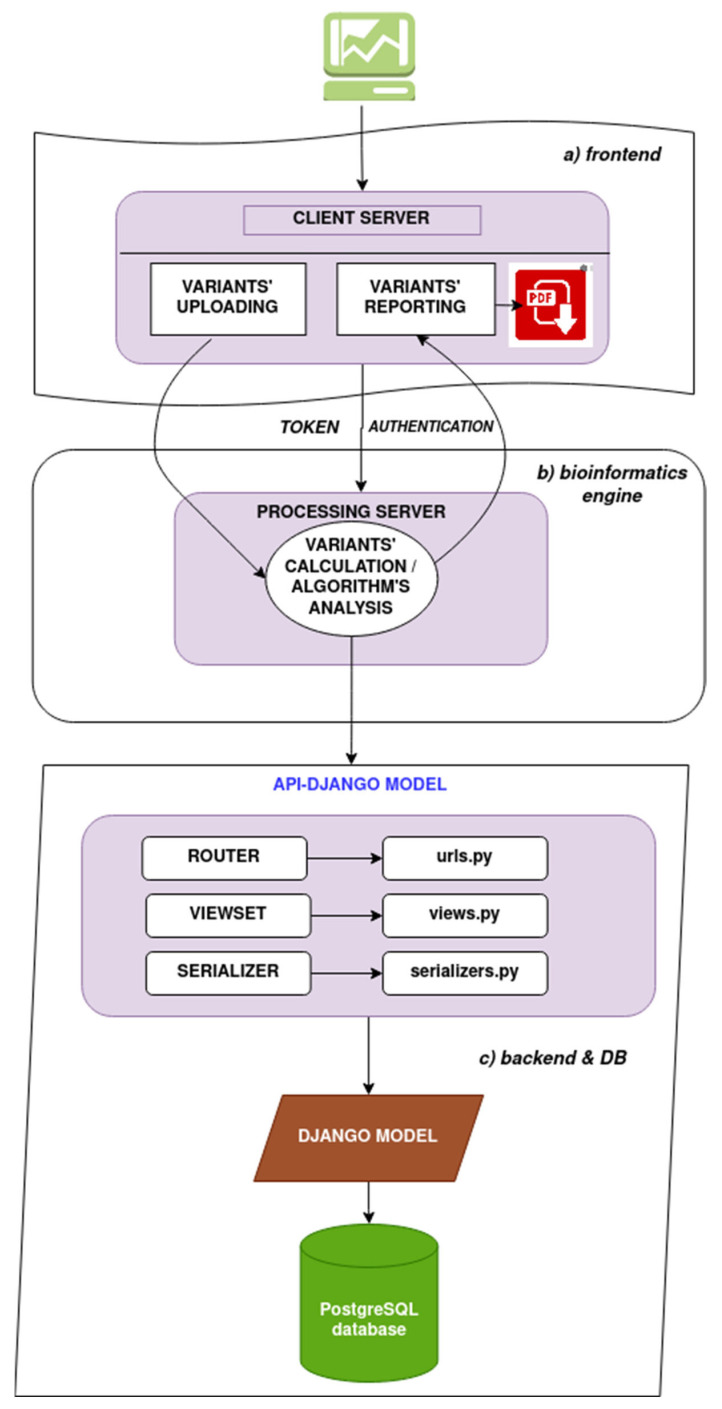
The general architecture of MAGI-ACMG web application. (**a**) The front-end section is built using Vue.Js. In this part, the user uploads the variants to be analyzed. The user can interact with the checklist to modify criteria and recalculate the final interpretation. The user can generate a PDF for each sample containing information on the analyzed variants. (**b**) Every query to the server has to pass an authentication mechanism. In this part, the bioinformatics engine performs the analysis for the uploaded variants and the MAGI-ACMG algorithm runs. (**c**) The back end is developed using Django REST Framework and PostgreSQL database. The data of the application is exposed as JSON via Django REST Framework application program interfaces (APIs). We have implemented serializers for converting data to execute requests and routing for API endpoints.

**Figure 2 genes-14-01600-f002:**
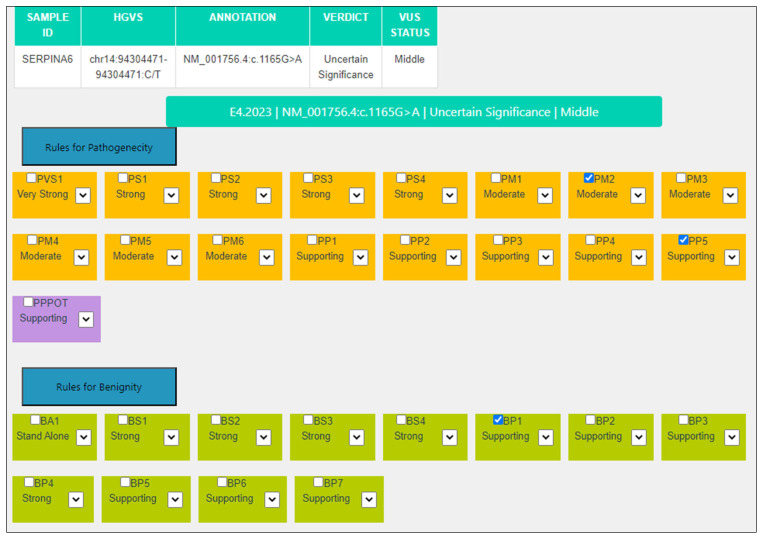
MAGI-ACMG web application screenshot for the *SERPINA6* NM_001756.4:c.1165G>A variant.

**Figure 3 genes-14-01600-f003:**
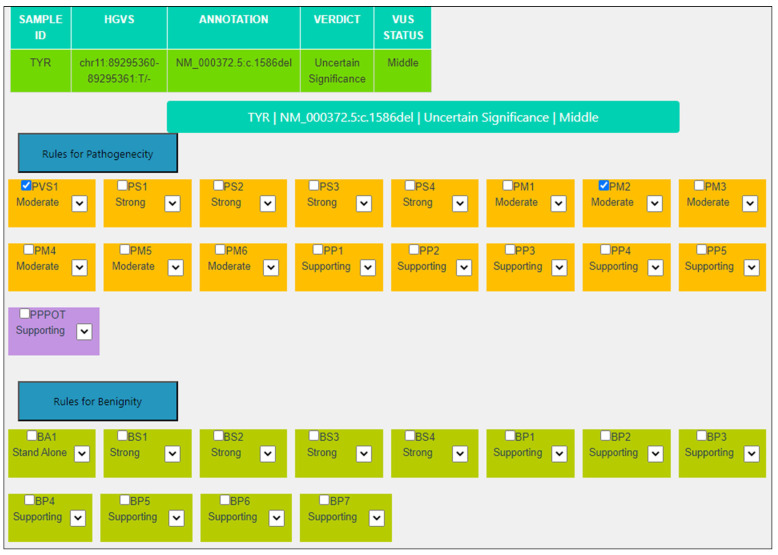
MAGI-ACMG web application screenshot for the *TYR*:NM_000372.5:c.1586del variant.

**Figure 4 genes-14-01600-f004:**
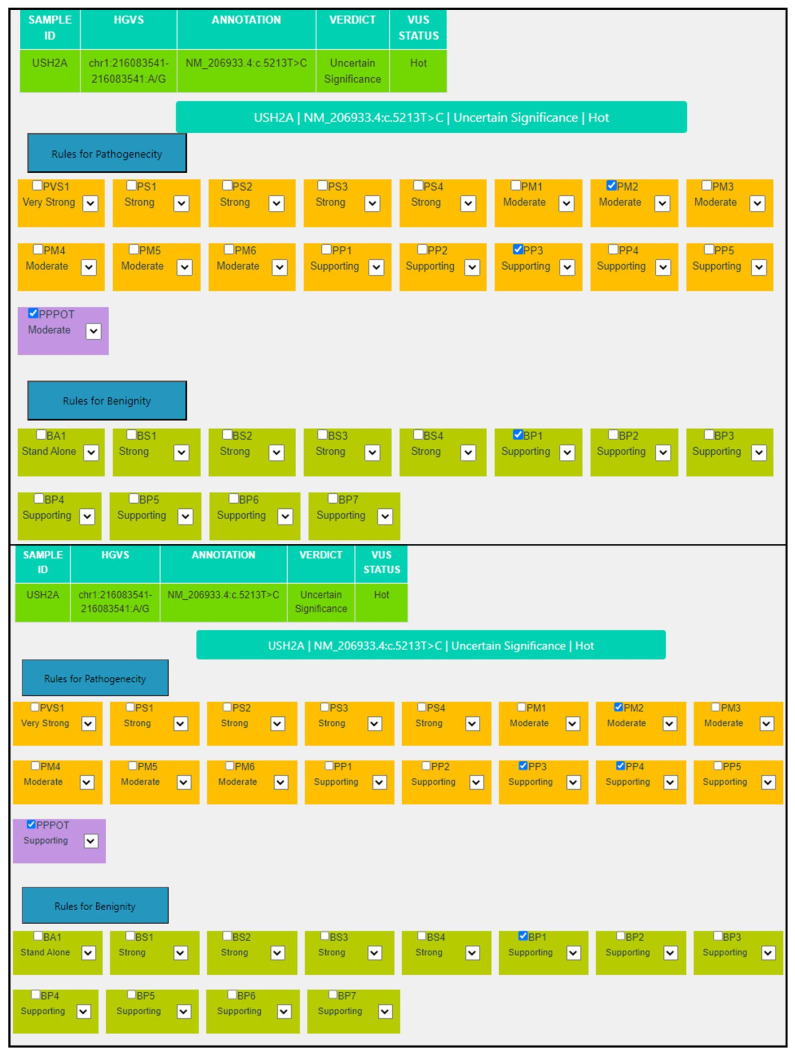
MAGI-ACMG web application screenshot for the *USH2A*:NM_206933.4:c.5213T>C variant.

**Figure 5 genes-14-01600-f005:**
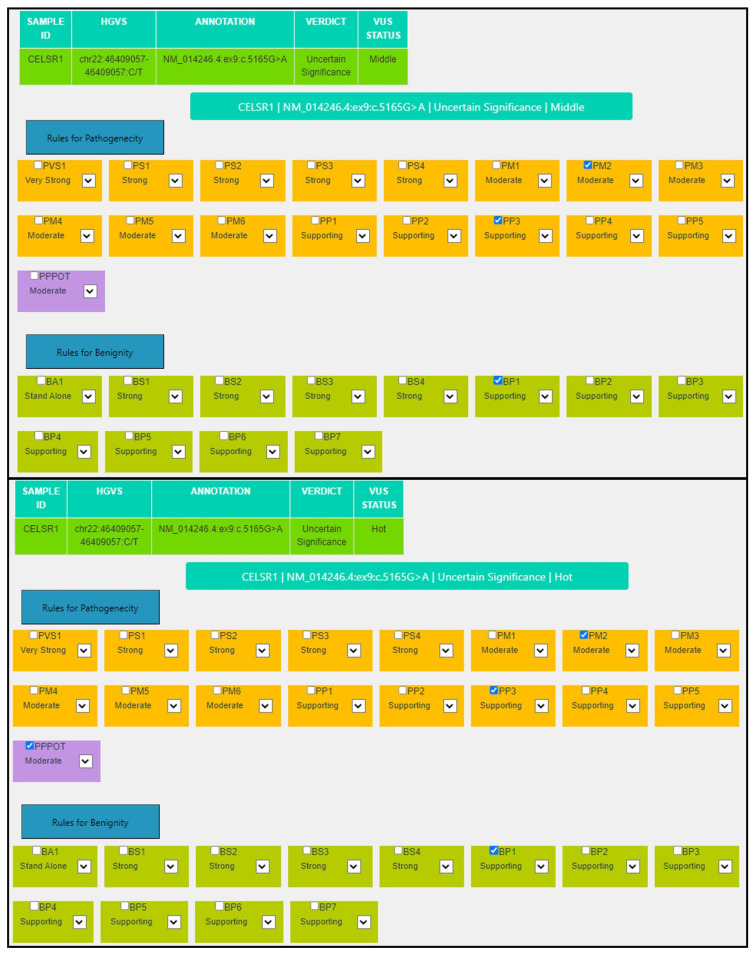
MAGI-ACMG web application screenshot for the *CELSR1*:NM_014246.4:c.5165G>A variant.

**Table 1 genes-14-01600-t001:** Subclassification of variants of uncertain significance (VUS) and combination of criteria that define them in each subcategory. In the diagnostic settings, Cold and Middle VUS are not reported. Dark blue: BS (Benign Strong); Light blue: BP (Benign Supporting); Green: PP (Pathogenic Supporting); Orange: PM (Pathogenic Moderate); Red: PS (Pathogenic Strong); Dark red: PVS (Pathogenic Very Strong).

VUS Class	Criteria Combination				
**HOT**					PVS
				BP	PVS
				PP	PVS
					PS
				PP	PS
			BP	PP	PS
			BS	PP	PS
				BP	PS
			PP	PM	PM
		BP	PP	PM	PM
		BS	PP	PM	PM
		PP	PP	PP	PM
	BP	PP	PP	PP	PM
	BS	PP	PP	PP	PM
		PP	PP	PP	PP
	BP	PP	PP	PP	PP
**MIDDLE**				PM	PM
			BP	PM	PM
				PP	PM
			BP	PP	PM
			PP	PP	PM
		BP	PP	PP	PM
				BS	PVS
				BS	PS
			PP	PP	PP
		BP	PP	PP	PP
**COLD**					PM
				BP	PM
				PP	PP
			BP	PP	PP
			BS	PP	PP
					PP
				BP	PP
				BS	PP
					BP
					BS

**Table 2 genes-14-01600-t002:** Comparison of criteria assigned through the VarSome stable-API v.11.6.1 system and MAGI-ACMG algorithm for the *SERPINA6*:NM_001756.4:c.1165G>A variant.

Criterion	VarSome Stable-API v.11.6.1	MAGI-ACMG
**BP4**	**Strong**: MetaRNN = 0.0134 is between 0.00692 and 0.108.	**Not applicable**: 11/15 predictors are damaging, REVEL score: 0.518 (>0.29) and CADD score is 24.78 (>22.7).
**PP3**	**Not applicable**	**Not applicable**: 11/15 predictors are damaging but REVEL score: 0.518 (<0.644) and CADD score is 24.78 (<25.3).
**BP1**	**Supporting**: GnomAD missense Z-score for gene SERPINA6 is −0.662 which is less than 2.99.	**Supporting**: GnomAD missense Z-score for gene *SERPINA6* is -0.662, which is less than 2.99.
**PM2**	**Supporting**: GnomAD genomes homozygous allele count = 0 is less than 2 for AD/AR gene *SERPINA6*, good gnomAD genomes coverage = 30.9.	**Moderate**: Absent from controls (or at extremely low frequency if recessive) in Exome Sequencing Project, 1000 Genomes Project, or Exome Aggregation Consortium.
**PP5**	**Supporting**: ClinVar classifies this variant as Uncertain Significance, 2 stars (multiple consistent, reviewed Apr ‘22, 5 submissions), citing 3 articles (PUBMED:17245537, PUBMED:12780753, PUBMED:10634411), associated with Corticosteroid-Binding Globulin Deficiency, with 5 submissions (1 P, 2 LP and 2 VUS).	**Supporting**: ClinVar classifies this variant as Uncertain. Significance, 2 stars (multiple consistent, reviewed Apr ‘22, 5 submissions), citing 3 articles (PUBMED:17245537, PUBMED:12780753, PUBMED:10634411), associated with Corticosteroid-Binding Globulin Deficiency, with 5 submissions (1 P, 2 LP and 2 VUS).
**Final**	**VUS**	**VUS Middle**

**Table 3 genes-14-01600-t003:** Comparison of criteria assigned through the VarSome stable-API v.11.6.1 system and MAGI-ACMG algorithm for the *TYR:* NM_000372.5:c.1586del variant.

Criterion	VarSome Stable-API v.11.6.1	MAGI-ACMG
**PVS1**	**Strong**: Null variant (frame-shift) in gene TYR, not predicted to cause NMD. Loss-of-function is a known mechanism of disease (gene has 78 reported pathogenic LOF variants). The truncated region contains 0 pathogenic variants. It removes 0.19% of the protein.	**Moderate**: NP6—LoF variants in this exon are not frequent in the general population and exon is present in biologically relevant transcript(s)—Variants remove <10% of protein.
**PM2**	**Supporting**: Variant not found in gnomAD genomes, good gnomAD genomes coverage = 31.3.	**Moderate**: Absent from controls (or at extremely low frequency if recessive) in Exome Sequencing Project, 1000 Genomes Project, or Exome Aggregation Consortium.
**Final**	**VUS**	**VUS Middle**

**Table 4 genes-14-01600-t004:** Comparison of criteria assigned through the VarSome stable-API v.11.6.1 system and MAGI-ACMG algorithm for the *USH2A*:NM_206933.4:c.5213T>C variant.

Criterion	VarSome Stable-API v.11.6.1	MAGI-ACMG
**PP3**	**Moderate**: MetaRNN = 0.859 is between 0.841 and 0.939, moderate pathogenic.	**Supporting**: 14/15 predictors are damaging, REVEL SCORE: 0.767 (>0.644) and CADD score: 28.30 (>25.3).
**BP1**	**Supporting**: 196 out of 529 non-VUS missense variants in gene *USH2A* are benign = 37.1% which is more than threshold of 33.1%.	**Supporting**: 196 out of 529 non-VUS missense variants in gene *USH2A* are benign = 37.1%, which is more than threshold of 33.1%.
**PM2**	**Supporting**: Variant not found in gnomAD genomes, good gnomAD genomes coverage = 31.0.	**Moderate**: Absent from controls (or at extremely low frequency if recessive) in Exome Sequencing Project, 1000 Genomes Project, or Exome Aggregation Consortium.
**PP4**	**Not applicable** (case specific).	**Applied**
**P-Pot**		**Supporting**: possible in-trans configuration with the pathogenic *USH2A*:NM_206933.4:c.2299del variant.
**Final**	**VUS**	**VUS Hot**

**Table 5 genes-14-01600-t005:** Comparison of criteria assigned through the VarSome stable-API v.11.6.1 system and MAGI-ACMG algorithm for the *CELSR1:* NM_014246.4:c.5165G>A variant.

Criterion	VarSome Stable-API v.11.6.1	MAGI-ACMG
**PP3**	**Moderate**: MetaRNN = 0.92 is between 0.841 and 0.939 ⇒ moderate pathogenic.	**Supporting**: 13/15 predictors are damaging, Revel score: 0.723 (>0.644) and CADD score: 29.1 (>25.3).
**BP1**	**Supporting**: 33 out of 33 non-VUS missense variants in gene CELSR1 are benign = 100.0% which is more than threshold of 33.1%.	**Supporting**: 33 out of 33 non-VUS missense variants in gene CELSR1 are benign = 100.0%, which is more than threshold of 33.1%.
**PM2**	**Supporting**: GnomAD genomes allele count = 2 is less than 5 for AD gene CELSR1, good gnomAD genomes coverage = 31.9	**Moderate**: Absent from controls (or at extremely low frequency if recessive) in Exome Sequencing Project, 1000 Genomes Project, or Exome Aggregation Consortium.
**P-Pot**		**Moderate**: possible de novo variant.
**Final**	**VUS**	**VUS Hot**

## Data Availability

The data presented in this study are available on request from the corresponding author.

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
