# Peer review of "MAGI-ACMG: Algorithm for the Classification of Variants According to ACMG and ACGS Recommendations"

_genes, 2023, doi:10.3390/genes14081600_

Round 1

Reviewer 1 Report

Since the algorithm is heuristic-based, it is difficult to understand why certain cutoffs were used without referencing literature

Screenshots show brightly colored boxes but text hard to read

Fig1 has no legend for colors

Author Response

Dear Reviewers,

Thank you very much for your attention in reading our manuscript and in your helpful comments and suggestions. We tried our best to answer all your comments, they were really helpful in increasing the overall quality of our manuscript.

We remain available if any other changing is needed. Best regards,

Gabriele Bonetti

On behalf of all the authors

Reviewer 1

Since the algorithm is heuristic-based, it is difficult to understand why certain cutoffs were used without referencing literature

We explained the used cutoffs with references in lines 140-143. In particular, we wrote “REVEL and CADD optimal cutoffs were retrieved from the recent ClinGen calibration study for computational predictors [4]. The REVEL score cutoffs (0.644 and 0.29) were applied also to the other predictors rankscores retrieved from dbNSFP v.4 [14].

Splicing cutoffs were decided referencing to literature: “AdaBoost and RF score cutoffs were retrieved from [15].” As it can be checked in literature the score is 0.515 and not 0.598, as previously reported, thus we modified it accordingly.

Screenshots show brightly colored boxes but text hard to read

Thank you for your comment. We agree that the screenshots were difficult to read. Thus, we cut the unnecessary sections of figures 2,3,4 and 5, and we modified the corresponding tables to report criteria details.

Fig1 has no legend for colors

Dear reviewer, we agree that a legend for colors is missing, but we think you are referring to table 1, and not figure 1. We think that in figure 1 a legend for color is not needed. Thus, we wrote the legend for colors for table 1 in lines 184-186.

Reviewer 2

In this work, Cristofoli and colleagues presented a new software for the classification of genetic variants. The authors integrate published methods including Varsome, AutoPVS1, and et al., with customized ACMG criteria to determine the classification. The method is interesting, yet some improvements could be made.

  1. In the introduction section, the authors should introduce more about the idea and current status of variants annotation. The author should also briefly describe the available methods and potential drawbacks of current methods. For instance, although this manuscript is heavily dependent on Varsome, the authors didn’t even cite the paper.

Dear, thank you very much for your comment. We agreed that the introduction section needed to be expanded, thus we added new paragraphs (lines 39-64).

  1. The authors selected the variants with MAF < 3%. Why do the authors choose this criterion? 

As reported in literature, we choose the MAF < 3 % because “Variants with a Minor Allele Frequency (MAF) below 3% are selected for annotation through VarSome Stable-API. The 3% threshold was chosen as a good compromise between 1%, which defines the term “polymorphism” versus “mutation”, and the computational burden of analyzing hundreds of variants, most of which are neutral.” This concept is reported in “F. Cristofoli et al., “Variant Selection and Interpretation: An Example of Modified VarSome Classifier of ACMG Guidelines in the Diagnostic Setting,” Genes (Basel), vol. 12, no. 12, p. 1885, Nov. 2021, doi: 10.3390/genes12121885.”, which we also cited in line 102.

  1. The authors only showed several examples. It’s not clear how different the overall result is similar or different from Varsome or other methods. The authors should at least perform a benchmark for a subset of variants.

We agree that a benchmark was needed to show clearly how the overall result is similar to Varsome. Thus, we added as Supplementary Table 1 the benchmark for a subset of variants. As also written in the discussion (lines 296-299), “the MAGI-ACMG classifier performs as VarSome when evaluating P, LP and VUS variants. However, the MAGI-ACMG algorithm allows VUS variants to be subclassified into the three categories Cold, Middle and Hot.”

Reviewer 3

The authors have proposed a classification scheme to select reportable variants with unknown significance by the proximity to either likely pathogenic/pathogenic or likely benign/benign. The core of the classification is a combinatorial scheme based on multiple scores, criteria, and cutoffs that frequently used to assess variant confidence/impact/etc. This is a very interesting topic and the website could be very useful when many variants have unknown significance, which happens frequently on novel or rare tumor subtypes.

My major comment is that the result section didn’t show any performance evaluation of the algorithm. I expected to see, for example, to predict the category of variations with known significance and evaluate how close the prediction is to the truth. The evaluation will support the effectiveness of the algorithm, and know whether the scores are combined in an appropriate way in the scheme, and how tight/loose the predictions are. 

We agree that a benchmark was needed to show clearly how the overall result is similar to Varsome. Thus, we added as Supplementary Table 1 the benchmark for a subset of variants. As also written in the discussion (lines 296-299), “the MAGI-ACMG classifier performs as VarSome when evaluating P, LP and VUS variants. However, the MAGI-ACMG algorithm allows VUS variants to be subclassified into the three categories Cold, Middle and Hot.”

Other comments:

All score types used in the combinatorial scheme need reference.

We explained the used cutoffs with references in lines 140-143. In particular, we wrote “REVEL and CADD optimal cutoffs were retrieved from the recent ClinGen calibration study for computational predictors [4]. The REVEL score cutoffs (0.644 and 0.29) were applied also to the other predictors rankscores retrieved from dbNSFP v.4 [14].

Splicing cutoffs were decided referencing to literature: “AdaBoost and RF score cutoffs were retrieved from [15].” As it can be checked in literature the score is 0.515 and not 0.598, as previously reported, thus we modified it accordingly.

Reviewer 2 Report

In this work, Cristofoli and colleagues presented a new software for the classification of genetic variants. The authors integrate published methods including Varsome, AutoPVS1, and et al., with customized ACMG criteria to determine the classification. The method is interesting, yet some improvements could be made.

  1. In the introduction section, the authors should introduce more about the idea and current status of variants annotation. The author should also briefly describe the available methods and potential drawbacks of current methods. For instance, although this manuscript is heavily dependent on Varsome, the authors didn’t even cite the paper.

  2. The authors selected the variants with MAF < 3%. Why do the authors choose this criterion? 

  3. The authors only showed several examples. It’s not clear how different the overall result is similar or different from Varsome or other methods. The authors should at least perform a benchmark for a subset of variants.

Author Response

(The authors gave the same response as above.)

Reviewer 3 Report

The authors have proposed a classification scheme to select reportable variants with unknown significance by the proximity to either likely pathogenic/pathogenic or likely benign/benign. The core of the classification is a combinatorial scheme based on multiple scores, criteria, and cutoffs that frequently used to assess variant confidence/impact/etc. This is a very interesting topic and the website could be very useful when many variants have unknown significance, which happens frequently on novel or rare tumor subtypes.

My major comment is that the result section didn’t show any performance evaluation of the algorithm. I expected to see, for example, to predict the category of variations with known significance and evaluate how close the prediction is to the truth. The evaluation will support the effectiveness of the algorithm, and know whether the scores are combined in an appropriate way in the scheme, and how tight/loose the predictions are. 

Other comments:

All score types used in the combinatorial scheme need reference.

Author Response

(The authors gave the same response as above.)

Round 2

Reviewer 2 Report

The manuscript is much improved now

Author Response

Reviewer 2

The manuscript is much improved now

Dear reviewer,

Thank you very much for your comment. We hope that our manuscript will be considered suitable for publication in your journal.

Best regards,

Gabriele Bonetti

Reviewer 3 Report

Thanks the authors for replying and adding the supplementary material. The supplementary file doesn't show sufficient evidence of the performance of the algorithm. To benchmark it, a specific design is needed, such as a large number of testing variations of all categories, define the true category of variations, and the statistics of the prediction performance (accuracy, F1 score, AUC, or others). Paragraphs in results and methods is needed to discuss the design and performance.

Author Response

Thanks the authors for replying and adding the supplementary material. The supplementary file doesn't show sufficient evidence of the performance of the algorithm. To benchmark it, a specific design is needed, such as a large number of testing variations of all categories, define the true category of variations, and the statistics of the prediction performance (accuracy, F1 score, AUC, or others). Paragraphs in results and methods is needed to discuss the design and performance.

Dear reviewer,

Thank you very much for your comment. We think that there is a misunderstanding about the MAGI-ACMG classifier and the aim of our manuscript, thus we will try to explain it better in the following paragraphs.

Our classification algorithm uses information retrieved through the VarSome Application Programming Interface (API). Our algorithm and how it works was already explained in the following articles:

  • Sorrentino, E et al. “Integration of VarSome API in an existing bioinformatic pipeline for automated ACMG interpretation of clinical variants.” European review for medical and pharmacological sciences 25,1 Suppl (2021): 1-6. doi:10.26355/eurrev_202112_27325
  • Cristofoli, Francesca et al. “Variant Selection and Interpretation: An Example of Modified VarSome Classifier of ACMG Guidelines in the Diagnostic Setting.” Genes vol. 12,12 1885. 25 Nov. 2021, doi:10.3390/genes12121885

The main focus of our manuscript is that our MAGI-ACMG classifier sub-classifies variants of uncertain significance (VUS) in the three classes “hot”, “middle” and “cold”. As we wrote in the manuscript (lines 73-77), the MAGI-ACMG classifier is specifically valuable because “the Association for Clinical Genomic Science (ACGS) suggested that a sub-classification system for variants of unknown significance (VUSs) might be useful for laboratories to decide which of these should be reported, according to the different levels of evidence supporting their pathogenicity and according to the likelihood that further data might allow a reclassification of variants as likely pathogenic or pathogenic [12].”
To our knowledge, the sub-classification proposed in this manuscript is not used by any other algorithm; thus, as no other tool is able to subclassify VUS, a benchmark is not possible.

Thanking you again for your hard work in reading and evaluating our manuscript, we hope that our manuscript will be considered suitable for publication in your journal.

Best regards,

Gabriele Bonetti